# Exploring the Impact of a Vicarious Learning Approach on Student Pharmacists’ Professional Identity Formation Using a Simulated Pharmacist–Patient Encounter

**DOI:** 10.3390/pharmacy11060177

**Published:** 2023-11-16

**Authors:** Laurie L. Briceland, Courtney Dudla, Alexandra Watson, Paul Denvir

**Affiliations:** 1Department of Pharmacy Practice, Albany College of Pharmacy and Health Sciences, Albany, NY 12208, USA; alexandra.watson@acphs.edu; 2Community Care Physicians, Latham, NY 12110, USA; cdudla@communitycare.com; 3Department of Allied Health Sciences, Albany College of Pharmacy and Health Sciences, Albany, NY 12208, USA; paul.denvir@acphs.edu

**Keywords:** empathic communication, observer role, pharmacist–patient communication, pharmacy education, positive and negative role-modeling, professional identity formation, role-playing simulation, vicarious learning

## Abstract

Purposefully developed professional identity formation (PIF) learning activities within the didactic curriculum provide crucial groundwork to complement PIF within authentic settings. The aim of this didactic exercise was to explore the impact upon student pharmacists’ PIF after viewing, analyzing, and reflecting upon a simulated pharmacist–patient encounter (PPE). A 12 min role-play video was created, featuring a pharmacist counseling a standardized patient on a new medication regimen; foundational principles of medication safety, health literacy, social determinants of health, empathic communication, and motivational interviewing were included in the counseling, with some aspects intentionally performed well, others in need of improvement. Also included were the patient’s varied reactions to the counseling. Students assumed the observer role and learned vicariously through viewing the PPE. Postactivity debriefs included justifying a foundational principle performed well by the pharmacist, and another in need of improvement, and a self-reflection essay expressing the impact of viewing the PPE on their PIF, from which extracts were thematically analyzed for impact. The main themes of the impact included increased awareness of counseling techniques, patient-friendly medical jargon, patient perspectives/empathy, positive and negative pharmacist role-modeling, and the value of the observer role. This PPE exercise enhanced PIF in terms of students thinking, acting, and feeling like a pharmacist, based on students’ self-reflections, which most often referenced effective pharmacist–patient communication and enacting optimal patient care.

## 1. Introduction

### 1.1. Professional Identity Formation (PIF) in Pharmacy

Professional identity formation (PIF) is an evolutionary and iterative process that pharmacy students begin upon matriculation in the program. Over time, through inculcation of practice roles in varied settings, students internalize the profession’s core values and beliefs [1]. The American Association of Colleges of Pharmacy (AACP) recently endorsed a policy statement: “AACP encourages colleges and schools of pharmacy to advance education that is aimed at the intentional formation of professional identity (i.e., thinking, feeling and acting like a pharmacist) …” [2]. Further, the revised Center for the Advancement of Pharmacy Education (CAPE) Educational Outcomes, known as COEPA 2022, includes this relatively new term ‘PIF’, which is emphasized in the Attitudes: Self-awareness domain [3]. Within the curriculum, PIF should be introduced early, continued longitudinally, and assessed at multiple junctures [4,5,6,7,8]. Scoping reviews in pharmacy and nursing education [9,10], as well as the AACP policy statement report [2], stipulate that educators should support the advancement of students’ PIF in multiple dimensions of pharmacy education, including didactic, experiential, and cocurriculum.

### 1.2. PIF Exercises in Didactic Pharmacy Curricula with Focus on Capstone Placement

Immersion into authentic practice experiences (i.e., experiential training, pharmacy workplace [10,11,12], and cocurriculum [13,14,15]) have been the hallmarks of PIF. Far more challenging, yet crucial to laying groundwork and complementing PIF in authentic settings, is the incorporation of meaningful PIF exercises within the classroom, in the didactic component of the curriculum. The challenge lies in the fact that pharmacy students do not perceive many didactic experiences to be impactful in supporting PIF [10]. Students attribute this perception to curricular emphasis on content acquisition instead of the desired active participation that would allow the application of skills and/or knowledge. Within a didactic learning activity, students express the desire for the inclusion of patient-centric authenticity, patient-facing positive role-modeling, and feedback on their classroom performance [10,12,13]. Thus, when designing PIF exercises for placement within classroom experiences, educators should incorporate patient-centric, provocative learning activities and assessments which purposefully afford students the opportunity to engage with pharmacists who are role-modeling professional capabilities [1,5,10]. Indeed, positive role-modeling by pharmacy practitioners and educators is essential to the development of a strong professional identity, as modeling can corroborate a student’s existing identity, provide inspiration, and predict future practice behaviors [2,6,10,16,17]. To advance PIF, exercises should be aligned with authentic pharmacy practice to minimize the disparity between what is taught in the classroom and what is practiced in the workplace [2,7,10,18]. Further, educators should explicitly discuss with students how the learning activities contribute to PIF, facilitating students’ appreciation of the value of the exercise [19,20].

A recent AACP report, Pathway to Professional Identity Formation, states: “PIF initiatives can find a fit within a nearly endless array of possibilities from orientation day to graduation.” [1]. Noble [10] indicates that the sequencing of PIF learning experiences is important to provide learners the opportunity to reconcile dissonance that might be encountered in curricular transitions (e.g., when students prepare for or return from experiential placements). The curricular transition from the final didactic semester of the third professional year (P3) into the Advanced Pharmacy Practice Experiences (APPEs) presents an opportune juncture to assess student pharmacists’ PIF and provide meaningful feedback/role-modeling to students. Many pharmacy curricula include capstone courses in the final semester of P3, primarily to assess pre-APPE skills and knowledge [21,22,23]. A multidisciplinary empirical study that assessed the purposes of capstone courses reported that, in addition to advancing skills and knowledge, capstone courses are employed to prepare students “for the next stage of the student journey”, such as entering the professional workforce or traineeships, and to develop students’ personal attributes [24]. These attributes have been embedded in capstone courses in the disciplines of law and business, and strongly underpin the development of a professional identity [24]. To our knowledge, there are no published studies within pharmacy education examining PIF in pre-APPE capstone courses, at that critical juncture of preparing students to enter the professional workforce as a P4 student pharmacist; we aimed to fill that gap.

### 1.3. Developing Awareness of Pharmacist–Patient Communication Skills Via Simulated Role-Play

Learning how to effectively provide patient-centered communication using the Pharmacists’ Patient Care Process (PPCP) is a foundational skill for student pharmacists [25,26]. A recent study involving P3 students reported that, of the six practice skills evaluated for pre-APPE readiness, the “green light” for patient-counseling competency was noted in less than 50% of students [27]. This finding indicates that continued efforts in developing pharmacist–patient communication skills are warranted, as studies demonstrate improved patient outcomes when provider –patient communication is optimized [28,29,30]. Further, student pharmacists value their roles as healthcare providers, and view patient-centered interactions and the development of ongoing patient relationships as highly influential in constructing their PIF [18].

Essential elements to be honed within patient-centric communication domains include: (i) health literacy/medication adherence [30,31,32,33]; (ii) medication safety [34]; (iii) social determinants of health/cultural competency [35,36,37]; (iv) empathic communication, including motivational interviewing themes of respect for patient agency/autonomy and shared decision-making [30,38,39,40]. To cultivate these necessary provider–patient communication skills in the health professions, a frequently used educational pedagogy is simulated role-play [41,42,43,44,45]. Role-playing methodology mimics authentic experience by employing critical aspects of the clinical scenario and enables the learner to practice in a safe environment without risk of patient harm [41,42,43,45]. The overarching goal of role-playing is to transfer learned skills into practice [46,47,48]. In role-play, typically the learner plays the role of the health professional, and the patient role is played by a “Standardized Patient” (SP), such as an actor, lay person, real patient, peer student [42,49], or faculty member [38]. There are myriad approaches to employing simulations in provider–patient communication, with studies demonstrating reasonable success with a variety of methods [41,44,45,50,51]. As some role-play methodologies can prove labor-intensive and expensive, for example, by employing paid SPs and/or requiring numerous faculty evaluators [42,52], a recent commentary on simulations argued that “educators should aim to just do enough in terms of prioritizing resource needs to ensure effective learning takes place but in such a way that resources are used appropriately.” [53]. For our PIF learning activity, we developed an innovative ‘low-resource’ simulation in which the students assumed the observer role in a pharmacist–patient role-play encounter.

### 1.4. Vicarious Learning through the Observer Role in Provider–Patient Communication Simulation

There is growing evidence that students can acquire communication skills and practice behaviors, typically acquired by firsthand participation, through observing the experience of others; this process is known as vicarious learning [54,55]. In the external role, the learner is listening/watching, but not directly participating in the simulated patient encounter [56]. To gain further benefit in the vicarious learning environment, the learner should receive observer tools (e.g., detailed explanations of the learning activities and an expectations checklist) and be engaged in postactivity debriefings, such as reflection to process the learning [56,57]. Incorporating guided self-reflection also happens to be an effective and frequently used evaluative means of influencing PIF, enabling students to meaningfully reflect upon the impact of the activity and to assimilate the transformative aspects of the experience into their own developing identities [5,9,13,16,58,59]. Considering the goal of PIF is to ‘think, act, and feel’ like a pharmacist, self-reflection enables the learner to demonstrate not only thinking like a pharmacist, but also feeling and acting like a pharmacist [5,13]. We developed an innovative PIF exercise within a pre-APPE capstone course in which our P3 students played the observer role in pharmacist–patient communication. Our project aim was to explore the impact of viewing, analyzing, and reflecting upon a simulated pharmacist–patient encounter on student pharmacists’ professional identity formation.

## 2. Methods

This pharmacist–patient encounter (PPE) simulation project was conducted at Albany College of Pharmacy and Health Sciences (ACPHS), a private college in New York which offers a traditional 4-year Doctor of Pharmacy program. The project underwent Institutional Review Board review and met the criteria for exemption from the requirements of federal regulations.

### 2.1. Curricular Preparation for Learning Activity and Capstone Course Context

During the spring 2023 semester, students completed this PPE–PIF exercise, described in Section 2.2, which focuses on pharmacist–patient communication. Students received instruction on pharmacist–patient communication over the three years of the pre-APPE curriculum, and practiced this skill in several courses, including the 6-semester Pharmacy Skills Lab sequence, P1 Foundations of Pharmacy, and summer Introductory Pharmacy Practice Experiences (IPPEs) in community and hospital settings. The PPE–PIF exercise was placed within a P3 course entitled Advanced Integrated Problem-solving Workshop (Advanced IPS), which serves as a required 2-credit pre-APPE capstone course. The course features clinical case work-ups incorporating simulated scenarios in which students apply the PPCP [25] and consider patients’ health literacy [60], medication adherence [33], and social determinants of health [36]. The course also includes activities such as protocol development of treatment guidelines and literature evaluation to help students identify with different pharmacist roles as a facet of PIF. Thus, this PPE–PIF exercise was intentionally placed at the end of the capstone Advanced IPS at a juncture where students are well-suited to apply their foundational knowledge, communication skill set, and professional roles/perspectives as they reflect upon their own PIF.

### 2.2. Description of the Learning Activity

The learning activity and instructional materials created for this PIF exercise focused on counseling aspects of pharmacist–patient communication (see Appendix A). The exercise was offered in an online asynchronous format, with the following objective presented to the students: “This PIF exercise was developed for the capstone Advanced IPS course as a method for you to demonstrate your PIF to date in terms of how you might think, act, and feel like a pharmacist when reacting to a Pharmacist-Patient Encounter.” The instructional materials informed learners that foundational pharmacy domains that could influence a patient’s medication adherence with a prescribed regimen such as: (i) medication safety/patient safety principles, (ii) health literacy principles, (iii) social determinants of health, and (iv) motivational interviewing/empathic communication strategies would be included in this PPE–PIF exercise. These foundational domains were taught and practiced in various pre-APPE coursework and experiences; intentionally, no new instructional material on these domains was included in the PPE–PIF exercise within the capstone course.

#### Instructional Material: Simulated Pharmacist–Patient Encounter (PPE) Video

The primary instructional material for this learning activity was a 12 min video of a simulated pharmacist–patient encounter (PPE) written and produced by three authors (see Document S2 “Script for Pharmacist-Patient Encounter PIF”). The encounter takes place in the Pharmacy Skills Lab counseling rooms on the ACPHS campus, adding an element of authenticity. The pharmacist is role-played by one of the pharmacist authors, and the standardized patient (SP) is role-played by a lay-person family member of a different author. The script included various reactions that the patient should express during the counseling session (verbally or by body language), including concern, confusion, hesitancy, as well as gratitude, relief, acceptance, and understanding, and was reviewed in detail with the SP prior to videorecording.

The focus of this PPE was a prevalent topic, diabetes education, in which the pharmacist counsels the patient on an injectable medication (Ozempic^®^) [61] to be added to the patient’s pre-existing diabetes regimen. All four of the foundational domains noted in Section 2.2 are interwoven within the 12 min video, in which the pharmacist performs most aspects of the foundational domains expertly. However, we intentionally included aspects within two domains (health literacy and motivational interviewing) that were medically accurate but in need of improvement in terms of provider–patient communication best practices [33,62]. The rationale for including both “right” and “in need of improvement” aspects was to present positive and negative role-modeling for the learner, intended to provide inspiration or provocation to which the learners would react [63]. Formulating judgments based on role-modeling aligns well with the tenets of PIF, as others have noted that students learn how to speak, act, and think like the health profession role-model they are observing [64].

Table 1 includes illustrative samplings of what is covered within the foundational domains in the PPE; see Document S2 Script for more information on the contents of the PPE.

Student learners in the course were instructed to view the PPE in the observer role. The observation tool provided to students was the instruction document for the assignment (see Appendix A), cueing learners to be on the lookout for checklist items: the pharmacist role-modeling the four foundational domains, and the patient’s reactions throughout the encounter. After viewing the PPE, students completed the written debrief, which included identifying one of the four foundational domains that the pharmacist performed well, and one that they did not perform well, with justification for each. The final component of the debrief assignment was the critical self-reflection, in which students crafted a narrative, reacting to the PPE with respect to PIF. Each student uploaded their assignment into the Learning Management System (LMS) by week’s end.

### 2.3. Assessment of Impact of Viewing the PPE on Students’ PIF

A mixed-methods design combining quantitative and qualitative data was employed. This included: (i) quantitative results of instructor-graded rubrics, (ii) frequency distribution of identified strengths and weaknesses of the four foundational domains covered in the PPE, and (iii) qualitative thematic analysis of student self-reflections. Additionally, upon reading the reflections, we counted how many students elected to write about a positive, inspiring aspect of the pharmacist’s counseling performance (positive role-modeling), or a provocative, negative aspect in need of improvement (negative role-modeling), and provided illustrative exemplars of stronger and weaker counseling formulations that students noted through the observer role. These data complement the quantitative data gathered through the rubric by providing rich context as to the actual impact of the PPE on students and informing instructors about how to enhance the assignment for future iterations.

#### 2.3.1. Instructor-Graded Rubric

An 8-point grading rubric was created to guide the assessment of the assignment. One instructor assessed all submissions and provided written feedback (reenforcing positive attributes and noting areas in need of improvement) to each student through the LMS. Students were assessed on both the critical self-reflection (4 points for sufficient critical analysis and application to future practice) and the selection/justification of aspects of the PPE performed well and in need of improvement (4 points). Rubric scores were collated from the LMS and are reported in Results. See Appendix A to view the rubric.

#### 2.3.2. Written Student Reflections

To gain a more nuanced first-hand perspective on this learning experience, students’ self-reflection papers were qualitatively analyzed. The prompt for the self-reflection follows: In 350–400 words, critically reflect upon your viewing of this Pharmacist-Patient Encounter (PPE). As a student pharmacist, what is the impact of this PPE upon you/your learning, either positively or negatively? As a result of reflecting upon this pharmacist’s component of the encounter (and its apparent impact on the patient), what are you now motivated to do (or do differently) in your (immediate or longer term) future when you encounter patients? To analyze the final reflection papers, we employed inductive thematic analysis, a qualitative method well-suited to finding patterns in textual data [66]. Induction is an iterative, “bottom-up” analytic process in which generalizations emerge out of repeated exposure to data, rather than through the application of a priori theoretical concepts or categories. Analysis proceeded in three waves, each supported by the qualitative analysis software Atlas.ti (Corvallis, OR, USA). In wave one, three authors read each reflection and entered free-text comments to identify recurring topics and themes of interest, known as open coding. Using insights from wave one, the authors developed an initial coding framework (a list of topics/themes) that could be applied systematically to the full data set. In wave two, the same authors identified relevant text extracts in each reflection and assigned all applicable coding categories to each extract, yielding 869 total extracts with an approximate length of two to four sentences each. Wave two was used to refine the initial codebook, eliminating codes that did not have adequate empirical support, combining codes that were revealed to share overlapping conceptual territory, and redefining code names to better convey the content and patterns in the data. The revised and final coding framework included 37 coding categories. In wave three, the same authors closely analyzed and discussed interpretations of the 20 coded collections that were most relevant to the current analysis: professional identity formation, pharmacist–patient communication, and observation of simulated patient-care encounters as a pedagogical technique for cultivating these qualities. This process yielded five main themes, as shown in Results.

The 4th author on our paper is a communication scholar formally trained in qualitative analysis and did not create or deliver this exercise. When considering reflexivity, the first three authors (who created and delivered the exercise) would inherently have reason to hope that the exercise was valuable and impactful; adding our fourth author provided balance when analyzing results.

## 3. Results

This PPE–PIF exercise was completed by 131 of 132 students in the Advanced IPS course, as one student was out on medical leave.

### 3.1. Instructor-Graded Rubrics

The average score on the grading rubric was 7.9 (out of 8.0) on the assignment, which demonstrates that the majority of students successfully developed a critical reflection and justified aspects of the encounter performed well and in need of improvement. Table 2 identifies foundational domains that students selected as examples of positive and negative role-modeling within the PPE, with illustrative exemplars for each.

### 3.2. Written Student Reflections

Table 3 presents the five analytic themes (with subthemes), the raw number of extracts that informed each theme, and an explanation with illustrative sample quotations, as derived from student reflections.

## 4. Discussion

### 4.1. Importance of Authentic Didactic Experiences to Support Professional Identity Formation (PIF)

The inclusion of PIF exercises within the didactic pharmacy curriculum is crucial to laying the foundation and advancing PIF in authentic practice settings [1,5,10]. Some authors have reported that pharmacy students do not perceive didactic experiences to be impactful in supporting PIF, attributed largely to the lack of patient-facing positive role-models and authenticity within the exercises [10,12,13]. We designed our didactic PIF pharmacist–patient encounter (PPE) exercise by featuring a patient-facing pharmacist modeling provider–patient communication in a realistic patient-centric setting; our students perceived this PPE positively for its realism, as shown in Table 3 (themes of impact). The purposeful placement of the PPE exercise at the end of a P3 capstone course was a unique feature of our PPE. Capstone courses are ideally suited to prepare students for transitioning into professional traineeships in the workplace (i.e., APPEs) and develop students’ personal attributes that serve to underpin the development of a professional identity, such as assuming responsibility, building confidence, and focusing on service to others [24]. Thus, we reasoned that placing the PPE in a capstone course provided an opportune juncture to assess students’ PIF, as students had completed the didactic and IPPE portions of the curriculum and were preparing to springboard into APPEs. How were our students assimilating into their roles as student pharmacists at this point in their journey? More specifically in PIF terms, how would they ‘think, act, and feel’ if they were the pharmacist in our PPE exercise? We elaborate upon these questions below in Section 4.1.1.

#### 4.1.1. Themes of Impact Derived from Simulated PPE and Contribution to PIF

As shown in Table 3, the main themes of impact that emerged from our qualitative analysis of 131 student reflections, listed in order of most to least prevalent, included an increased awareness of counseling techniques; patient-friendly medical jargon; patient perspectives/empathy; positive and negative pharmacist role-modeling; the value of the observer role in learning. Our PPE positioned students in the observer role, in which the learner is not directly participating in the simulated patient encounter [56]. As such, in our PPE, the student observed the roles of both the pharmacist and the standardized patient (SP), learning through a process known as vicarious learning [54,55]. To maximize benefit in the vicarious learning environment, we provided the learner observer tools (e.g., explanations of the learning activities and expectations) and a postactivity debriefing reflection exercise to process the learning [56,57]. We incorporated guided self-reflection as an effective and frequently used evaluative means of influencing PIF, enabling students to meaningfully reflect upon the impact of the activity, and assimilate transformative aspects of the experience into their own developing identities [5,9,13,16,58,59]. Our study adds to the body of literature demonstrating the importance of self-reflection in supporting PIF in didactic exercises.

We believe the two intertwined features of the observer role and ability to view both pharmacist and SP roles are unique features of our PPE within pharmacy education [45,67]. Importantly, in the observer role, our students were able to gain more awareness of the patient’s reactions to the pharmacist’s counseling, which opened students’ eyes to the patient’s perspective. Specifically, students could see on display the patient’s social cues, such as confusion, reluctance, worry, as well as gratitude, relief, and acceptance; they stated that this display served as motivation to be more attentive to patient’s social cues as they aim to incorporate empathic listening and communication into future practice. Further, in Table 3 quotations, students reported that, in other courses, they were assigned to actively role-play the pharmacist role; in that role, the student was concentrating on completing the requisite counseling tasks within a defined period and admitted to missing the SP’s social cues. By being able to observe a role-modeling pharmacist interacting with an SP, instead of themselves playing the pharmacist role, students were able to learn from both roles and gain important perspectives to carry forward into practice. Also shown in Table 3, students valued this PPE exercise in terms of its authentic patient-centricity and underpinning for their own PIF; in other words, how they viewed themselves thinking, acting, or feeling like a pharmacist.

#### 4.1.2. Positive and Negative Patient-Centric Role-Modeling to Support PIF

Role-modeling is an effective educational strategy which enables the learner to make judgments as to whether they would choose to emulate the observed behaviors [63]. Students are heavily influenced by practitioners who positively model patient-centered care and interactions [18,64,68], and our results corroborate this contention, which is especially evident in Table 2 quotations, where the pharmacist was expertly role-modeling medication safety and empathic communication. Specifically, students were sensitized to many important counseling tips (e.g., to show the patient the actual pen, to allow the patient to practice with the pen, and to show the patient how to dispose of the pen) and new approaches to actively listen to the patient’s concerns. Medical student educators explored using both positive and negative modeling in simulated role-play to teach clinical communication skills [51,63,69]; they determined that students valued being exposed to both positive and negative provider–patient interaction scenarios [63], and instinctively discerned which encounters were unsatisfactory [69]. Our PPE exercise, in which our pharmacist purposefully demonstrated both positive and negative role-modeling, another unique component of our exercise, tested and corroborated this premise that students would be able to readily discern unsatisfactory components of the encounter. As Table 2 depicts, 78% of students discerned our pharmacist’s negative role-modeling in terms of health literacy (use of complicated medical jargon), with an additional 19% noting improvements that were needed in motivational interviewing and SDOH domains. Further, the illustrative quotes in Table 3 provide testimony that the negative role-modeling provided sufficient provocation to motivate the student to not emulate a particular behavior.

Formulating judgments about role-modeled behavior aligns well with the tenets of PIF; research has shown that students learn how to speak, act, and think like the professional role-models they observe [64]. Such role-modeling by pharmacy practitioners is essential to PIF, as modeling can validate a student’s existing identity, provide inspiration, and predict future practice behaviors [2,6,10,16,17]. This exercise required students to articulate shortcomings in a pharmacist’s performance, cultivating both readiness and willingness to make these judgments as they transition into learning from preceptors in their upcoming APPEs. Preceptors are only human and will sometimes exhibit moments of negative role-modeling. It is important that student pharmacists feel capable of making internal judgements about what is worth emulating in their preceptors, mentors, and other role-models. Research indicates that students often experience dissonance between education and practice [18], which may be helped by didactic learning that prepares students to make sense of, and even learn from, experiences that do not conform to idealized expectations [10]. In these data, one of the more consistent patterns across multiple themes in Table 3 is how often students followed up their criticisms with better versions of what the pharmacist had tried to do. Based on these results, we believe that pairing negative role-modeling with critical reflection and analysis is a potentially powerful tool for developing PIF in didactic settings.

### 4.2. Student Perspectives on Optimizing Foundational Principles in Pharmacist–Patient Communication

Through participation in this PPE exercise, and as exemplified in the student quotations in Table 3, students recognized the relevance and benefit to future practice of optimizing pharmacist–patient communication skills. In our assignment instructions for students, we had stated: “Keep in mind that as a pharmacist, you may evaluate a patient’s pharmacotherapy regimen, and tweak the medications/dosage perfectly for the individual patient’s goals, but if the patient doesn’t take the medication regimen correctly or at all, then perhaps all your hard work on optimizing the medication regimen was for naught.” Indeed, in their own words through critical self-reflection, students expressed how this PPE opened their eyes to the importance of pharmacist–patient communication in terms of achieving a positive patient outcome. They noted that there is much more to an effective PPE than just providing the medication to the patient. Specifically, students identified the significance of effectively addressing the foundational principles we featured in the PPE, including medication safety, healthy literacy, social determinants of health, and motivational interviewing, during patient encounters. Further, students expressed that these principles should be incorporated into practice in a patient-centric manner, empowering each patient to have a voice in their care, ultimately leading to improved adherence and outcomes [46]. Students emphasized the impact of adapting counseling to patients’ health literacy levels to ensure understanding of the messaging and tailoring recommendations to patients’ preferences/goals.

By observing the pharmacist conduct the PPE, and by reflecting upon their own experiential and workplace experiences to date, students were able to formulate individualized assessments of techniques and behaviors they would like to emulate (or avoid) in their own practice. For instance, many students noted that thoroughly reviewing the patient’s current medications using medication visualization, and counseling on new medications using device demonstration (if applicable) and patient teach-back, would be excellent strategies to employ to positively influence adherence. Students offered transformative thinking, reflecting on ways they can improve in practice, even as a student pharmacist, and were excited to put these ideas/behaviors into practice immediately. Similarly, taking time with patients to actively listen to their concerns and effectively answer their questions, despite at times having limited resources to do so, will promote trust and rapport [40]. This exercise encouraged students to see the encounter from the patient’s perspective, which is the essential empathy that healthcare providers need to nurture in themselves [40]. Student reflections revealed multiple dimensions of empathy, including emotional, cognitive, and procedural aspects; they treated the patient’s fear and confusion as reasonable, and even articulated their concerns about insurance, health literacy, and SDOH barriers in ways that empathically acknowledged the varied backgrounds and experiences that patients bring to medical encounters.

### 4.3. Limitations and Future Directions

There are limitations to our work. This was an ungraded assignment, and thus students could have given it less than their best; though, anecdotally, the instructors were pleased with the students’ efforts on this assignment. Further, this was an online asynchronous assignment, so all correspondence (i.e., instructions and feedback) were provided in writing through the LMS; thus, while students reacted well to the assignment instructions (i.e., no issues with completing the assignment on time/accurately), they may not have read the written feedback that the instructor provided to each student, which was intended to close the loop on PIF development. The uncertainty of whether the student is reading the feedback is an inherent shortcoming of online communication in general [70], and methods to improve this process of providing feedback are warranted. Lastly, this work was but one assignment on PIF, not a longitudinal series over time, which could minimize the impact of the assignment on students’ PIF. Repetition of similar exercises over time would likely increase the PIF impact and provide a longitudinal view of PIF development.

Our innovation is readily transferable to other courses and to other pharmacy colleges. For our own future directions, in keeping with debriefing recommendations in the literature [56,57], we plan to provide the video script to students a priori, and develop a checklist for student use as they view the encounter. Also, Koponen [51] reported that if students discussed the exercise in groups, in debrief, they learn/reflect more on the problematic doctor–patient interaction components, which is an intriguing possibility for our next iteration.

## 5. Conclusions

Observing, analyzing, and reflecting on simulated pharmacist–patient encounters is a potentially impactful method of professional identity formation in didactic curricula. Simulations that intentionally include both positive and negative role-modeling may be particularly beneficial in preparing students to navigate dissonances between idealized expectations and the challenging realities of practice. Well-scripted and well-acted video portrayals of patient counseling create opportunities for students to observe and interpret patients’ social cues and body language, a key foundation for empathic communication in clinical settings.

## Figures and Tables

**Table 1 pharmacy-11-00177-t001:** Sampling of foundational domain content included in the pharmacist–patient encounter.

Foundational Domain Showcased in PPE	Illustrative Examples of What the Pharmacist Discussed in PPE
Medication-Safety/ Patient-Safety Principles	Five “Rights”: patient, drug, time, dose, and route [34] Thoroughly reviewed other medications that the patient was taking for diabetes Demonstration of the actual injectable Ozempic ^®^ pen, how to prime the pen, and allowed patient to practice using a demo set-up utilizing the teach-back method Information on how to dispose of needles and demonstration using actual sharps container Information on how to rotate the injection site on the body
Health Literacy to Improve Medication Adherence	Teach-back method of patient education to ensure that the patient was understanding the verbal instructions Pharmacist spoke too quickly during the contraindications section of the medication discussion Pharmacist spoke at grade level well above the recommended 5th–6th-grade level for the general population, using complex medical terms such as medullary thyroid carcinoma [60] Pharmacist summarily provided patient with package insert, which on average is written at a 10th-grade level [65]
Social Determinants of Health	Barriers that could preclude the patient from adhering to Ozempic®, such as affordability, were discussed Identification in advance of a coupon for the patient and confirmed insurance coverage would be reviewed Pharmacist did not complete a comprehensive review of SDOH that might affect this patient’s ability to adhere with regimen, including transportation issues and access to refrigeration in the home
Motivational Interviewing/ Empathic Communication Strategies	Pharmacist built a trustworthy patient–provider relationship throughout the encounter, expressing empathy Pharmacist authoritatively told the patient that the best course of action for her diabetes at this time was to use the injectable Ozempic® and did not allow the patient to develop discrepancy and have a shared voice in decision making Pharmacist did not always roll with resistance when patient was reluctant to take the new injectable medication Pharmacist did not always practice active listening

PPE = pharmacist–patient encounter. SDOH = social determinants of health.

**Table 2 pharmacy-11-00177-t002:** Students’ selection of positive and negative role-modeling exhibited by the pharmacist in the pharmacist–patient encounter.

Foundational Domain Exhibited in PPE	Positive Role-Modeling N (%)	Illustrative Examples	Negative Role-Modeling N (%)	Illustrative Examples
Medication Safety/ Patient Safety principles	72 (55)	“The pharmacist verified the patient’s name and birthday, asked open ended questions regarding the medication and looked directly at the medication in the bottles to ensure that the way the patient was taking the medication and the medication itself in the bottle matched the label.”	4 (3)	“The pharmacist was also quite rude to the patient, as seen with “correcting” the patient on the indication for atorvastatin, as well as reminding them they are on the maximum dose of metformin, and yet they are still doing a bad job managing their blood sugar.”
Health Literacy to Improve Medication Adherence	7 (5)	“I really liked how the pharmacist discussed A1C to the patient, like explaining what it is, what the target level is in diabetes, and complications that can arise for the patient if her A1C is not controlled.”	102 (78)	“The pharmacist could work on health literacy because they used complex medical terminology when describing the patient’s new medication. It is extremely important to use terms that patients know or would be able to understand.”
Social Determinants of Health	7 (5)	“The pharmacist mentioned the cost and assured the patient they would follow up with the pharmacy if authorization is needed. She also went over the savings card which can further reduce the cost with the copay.”	13 (10)	“The pharmacist mentioned that there were coupons available that she could use that could be found online […] not everyone is tech savvy, has the resources to look at things online or would know where to look this up.”
Motivational Interviewing/ Empathic Communication strategies	45 (34)	“The pharmacist did a good job gauging the patient’s comfort with the new suggested medication and used friendly and understanding language, building trust between the patient and pharmacist, allowing the patient more time to gain comfort, confidence, and understanding with the medication.”	12 (9)	“The pharmacist did not slow down to really listen to the patient’s concerns about using injectables and instead focused solely on providing information, neglecting the importance of empathic communication. This could cause the patient to feel unheard and unmotivated to adhere to their medication regimen.”
Total	131 (100)		131 (100)	

PPE = pharmacist–patient encounter.

**Table 3 pharmacy-11-00177-t003:** Themes derived from the analysis of student reflections.

Main Themes/(Subthemes) (N) *	Explanation	Sample Quotation
Value of video-based role-modeling (73) (Positive and Negative Role-modeling)	Students found value in taking on an observer role, prompting reflection on the encounter from both the pharmacist and patient perspective. Many indicated that watching another pharmacist conduct a patient encounter motivated them to incorporate what the pharmacist did well into their future practice as well as change what they felt could have been improved upon. Students easily recognized overt errors, such as inappropriate medical jargon, as well as subtler flaws, such as mishandling delivery of the package insert and gaps in empathy. Observing stronger and weaker moments of pharmacy counseling, including patient reactions to both, invited reflection about how to adapt the practices modeled in the video while working in the field.	“By watching this pharmacist-patient encounter, I was able to take an outside look at how a counseling session may actually happen. It is difficult to objectively judge the quality of an interaction when you are the one actively taking part in it. I speak with and counsel patients all the time at my job in community pharmacy but I’m not always actively thinking about if I’m using the most patient friendly language or instilling confidence and trust in me with my eye contact and body language. When I watched this video, I was able to focus on the way the pharmacist was communicating, not just what information she was specifically sharing. Seeing examples like this (and being able replay sections) helps me to be able to pick out strengths and weaknesses not only in the videos but in my own practice as well.”
Situating video-based role-modeling within past and future learning: Complexities of realism and idealism (50) (Teaching/learning communication training, realism, and outside perspective)	Students perceived this as a valuable pivot from past learning to future learning. The video provided an integrated demonstration of previous knowledge and skills (“bringing it all together”) as well as a preview of future experiential learning during upcoming APPEs. Reflections revealed a dialectic tension between realism and idealism. Many students praised the realism of the encounter, which provided a naturalistic portrayal of counseling and articulated an aspirational view that this is what counseling should look like in all practice settings. Others wondered if the portrayal was too idealistic given the time demands of a busy community pharmacy, lamenting that in-depth counseling may only be possible in clinical or hospital settings.	“Even though we have practiced plenty of patient encounters in [pharmacy] skills lab, when I practiced those, it was always with the lens of “what steps do I need to remember to do well on this assessment?” and I would tend to forget the patient might get lost in the details. So, it was refreshing to view a patient encounter from an outside perspective, allowing me to see what I liked and thought needed improvement from this different vantage point.”
Observing and evaluating counseling approaches and techniques (256) (Active listening, open-ended questions, and taking time with the patient; medication and device visualization; offered package insert)	Students demonstrated awareness of myriad specific counseling approaches and techniques portrayed in the video, often articulating how they might enhance or inhibit rapport depending on their execution. Many recognized the importance of more generalized approaches, such as active listening and taking time with patients to build their trust. They also evaluated specific communication techniques, such as open-ended questions, physical device demonstration, and inviting patient teach-back to foster patient confidence, safety, and adherence for improved outcomes.	“Many patients have low health literacy skills, which can lower their ability to understand medication instructions and take medications correctly. As a future pharmacist, I think that it is essential to use plain language and teach-back methods (with demos) to effectively communicate medication instructions and ensure patient understanding. By doing this, we put our patients in control of their own health education. Instead of talking at patients with instructions they do not understand, we can ensure equitable education for all patients for all medications.”
Medical jargon: Expertise and accessibility (147) (Patient-friendly versions of medical jargon)	Among the four foundational domains, excessive medical terminology (jargon) was the most cited error, a failure to adapt to patients’ health literacy. Many indicated that jargon led to patient confusion, evident through the patient’s nonverbal cues and multiple clarifying questions. Students noted that “spewing” hard-earned expertise can be tempting but ultimately threatens rapport and leads to lower adherence and subsequent poor outcomes. In some cases, students offered patient-friendly versions of jargon portrayed in the video, often framed as versions they plan to use when caring for their own patients in the future.	“A lot of times, as students, we get excited when we know an answer to a health or medication question, and we start regurgitating information we’ve learned in class not realizing that many patients do not have the background knowledge we do. I definitely have taken my knowledge for granted and I need to remind myself who I am speaking to with each conversation. No one ever likes being spoken to in such a way that they leave the conversation feeling worse off than when they started the conversation.”
Empathy: Seeing patient perspective (83) (Reading social cues/patient (dis)comfort)	The video promoted reflection about three aspects of empathy, all rooted in sensitivity to patient perspective: (1) emotional (treating fear/concern as understandable); (2) cognitive (adapting to lower health literacy or confusion); (3) practical (guiding patients through unfamiliar procedures). Emphasizing the unique value of observing video role-play, students routinely cited the patient performance—her body language and other social cues—as evidence for claims that the patient “looked, seemed, or appeared” nervous, skeptical, or overwhelmed. Empathy requires active listening and attention to social cues, and the video provided an opportunity to cultivate that sensitivity.	“Being an outside observer of this interaction helped me clearly see when the patient felt confused or uncomfortable, which is something that may not have been apparent to the pharmacist at that given moment in time. This made an impact on me because it made me realize that there may have been times that I did not perform optimally in a specific domain and may not have noticed at that time of interaction. This motivates me to try to look out more for cues like these when I am counseling patients and performing these encounters.”

* The number shown here refers to the number of extracts out of the total of 869 extracts that contained material relevant to the theme or subtheme.

## Data Availability

Data are contained within the article.

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
