# Peer review of "Exploring the Impact of a Vicarious Learning Approach on Student Pharmacists’ Professional Identity Formation Using a Simulated Pharmacist–Patient Encounter"

_pharmacy, 2023, doi:10.3390/pharmacy11060177_

Round 1

Reviewer 1 Report

Comments and Suggestions for Authors

Thank you for the opportunity to review your paper on the impact of vicarious learning on professional identity formation in pharmacy. PIF is a popular topic currently, hence there will be interest in your paper.  I have some suggestions for you to consider to improve your paper.

Overall, the paper is quite long and some of descriptions repeat in the introduction and conclusion. You may be able to be more concise in your descriptions to keep the reader engaged. I think you can also consider shortening some of the narratives in your results.

Also, you talk about didactic curriculum and that you focus on a PIF exercise within the didactic curriculum (section 1.1 line 52-53) however you then state that the exercise is in a course entitled Advanced Integrated Problem Solving Workshop....which does not sound entirely didactic? Also, your exercise has tools built in to make it an active learning observation, so not sure it actually fits the definition of didactic?  You may consider defining how you are defiing didactic curriculum and how your exercise fits? Or consider removing the didactic conversation.

Introduction: You state the focus of your research in section 1.1 line 52, then you state an aim at the end of section 1.2 line 98-100, then you state a slightly different project aim in section 1.4 line 153-155...and in between you allude to your PIF exercise...it is all a bit confusing to the reader. You may consider consolidating the purpose of your paper but clearly outlining the problem your paper is trying to solve? What is your goal/purpose and then clearly state the research questions your study set out to solve.

Methods:

You may consider noting the reason ethics was not required.

You may also describe the backgrounds of the researchers analyzing the data and the importance of that.

Results

You may consider condensing some of the narrative in some places...i.e. truncate some of the quotes to the relevant pieces only.

Discussion: In the limitations you may also consider discussing the impact of one assignment in one course on PIF....how much impact is realistic?

A section on reflexivity is important as you have a large qualitative component.

Author Response

See attached Response to Reviewers document. 

Reviewer 2 Report

Comments and Suggestions for Authors

I enjoyed reading this well written manuscript.  The authors have given due consideration to the study design and the impact that their study would have on the PIF of the students.  Overall, well done.

One minor comment, coming from a school that uses OSCEs (admittedly, expensive!).  Does your institution also use OSCEs?  or is this a low cost alternative?

Author Response

Please see Attached Responses for Reviewers document

Reviewer 3 Report

Comments and Suggestions for Authors

I found the study very interesting and topic is very well explained. I have some littele recommendations to authors.

From my point of view the section 'Methods" is very long and very difficult for the readers. If it is compressed will be clearer. I think that authors could add information about the total number of students in advanced IPS course. We understand that 132 participated, but total number is not written

In the methodology section is written that "The focus of this PPE was diabetes education in which the pharmacist counsels the patient on a new injectable medication (Ozempic® )"….It is not absolutely clear how this is implemented in current study exactly and why the authors focused on this example?

Author Response

(The authors gave the same response as above.)

Reviewer 4 Report

Comments and Suggestions for Authors

I have read the manuscript and here are some comments. The paper is of interest to people working within pharmacy education.

Table 1 needs to be improved to increase readability. Bullets are not suitable.

Did you not consider to make an evaluation afterwards to also gather the students' opinions of the exercise besides analysing their reflections and grading the assignment?

Some abbreviations are not defined in the paper for example SOAP.

To quantify the qualitative responses from the student reflections is not necessary. I would recommend to focus on the qualitative analysis.

Table 2 and 3 are difficult to read and needs to be improved. I would suggest not to summarise the qualitative analysis in a table but rather write this in the text. The sample quotations are long. I would suggest to shorten the quotations and include quotations from more students to each theme to illustrate the theme.

What kind of feedback were the students given on their assignment? Please clarify.

I believe that the suggestion to also include discussions in groups in the future would be beneficial to the students' learning experience. 

Author Response

(The authors gave the same response as above.)

Reviewer 5 Report

Comments and Suggestions for Authors

The authors have developed a video portraying positive and negative patient pharmacist interactions, which was followed by students analyzing the encounter and reflecting on positive and negative behaviors to hopefully apply to their own practices.  The video captured the social science aspects and behaviors vs. content.  Emphasis on communications, culture, SDOH, motivational interviewing is very important, which is not always emphasized in curriculums and capstone courses.  The paper is very well written and referenced.

Role-playing in a classroom with student discussion on the actors is not a new teaching pedagogy.  Granted self-reflection is not always a requirement, and not everyone publishes their student learning, so some newness from this manuscript.   This sort of exercise seems more important for early teaching and development of these skills than as a competency assessment prior to clinical rotations. 

Very true that capstone observation of student abilities is expensive and resource-intensive. Exploring different ways to document competency would be helpful.  This simulation does not document student competency but knowledge, and is a good review of best practices previously taught.   However only one positive and negative behavior was required for commentary, so we do not know if students could correctly identified all positive and negative behaviors and or if they would do all the positive behaviors in their patient interactions.

The missing link is will observation and scoring of another pharmacist patient interaction result in the students’ abilities to actually do patient encounters using best practices, which is a leap in assumptions.  Correlating simulation abilities to actual student competencies would have been a great project, and then justify replacement of resource-intensive capstone assessments.  

In terms of professional identify formation (PIF), I do not see this impacting any new aspects of the think, act, and feel paradigm. Although a new movement in pharmacy education, most of the things are what we would call clinical pharmacy skill advancement with new emphasis on the feeling components. Furthermore the students should have already developed the skills assessed earlier in their education journey or while on internships and IPPEs. Because no new practices were developed, students passively learned to collect patient data, and the exercise does not result in accountability, responsibility, nor advocacy, I do not feel this vicarious exercise is a PIF advancement exercise.

The project is more like a quality improvement assessment, or maybe thinking about students’ abilities to evaluate other students, esp with scaffolding teaching with P4 students teaching P1-P3 students.   

The evaluation can be a measure of vicarious learning, which you proved did result in identification of some positive and negative behaviors, but we do not know if any of it was new learning vs. review, and accuracy with identification of all negative behaviors.  

This paper does not prove that this simulation with reflection would be better than videoing actual student performance during capstone courses and then the students watching their video with your checklist to do a reflection on their own performance, which I would see as competency or accuracy vs. PIF.  

Your rubric has some value for others to use for these observation video reflections. 

The themes from the qualitative work are helpful but most of the quotes were not enlightening and take up much space.

Would have been nice to also include behavior models like the health belief model and change readiness model in your rubric, which are critical in changing patient behaviors to implement health lifestyles and medication use. 

Changing this paper to discussion of a vicarious learning exercise with reflection late in a pharmacy curriculum would be better than a PIF exercise.  Comparisons to other vicarious learning that is quite common in pharmacy curriculums would then be your discussion.  I still think the key benefit of this project could be your rubric, which could be added to the supplementary files.

Author Response

(The authors gave the same response as above.)

Round 2

Reviewer 1 Report

Comments and Suggestions for Authors

Thank you for your detailed responses to the reviewers. The edits to your manuscript improve it considerably. I have no further comments.

Reviewer 5 Report

Comments and Suggestions for Authors

The authors have added some clarifying content and added their rubric to the supplemental material.   In some places, the narrative has been minimized, but the manuscript is still long.  The authors still see this exercise as a major PIF exercise, which I don't, but they explain in detail in the responses to reviewers.  The feedback didn't change my mind, but always two perspectives to everything and or over time renaming things does happen.  With PIF in the title, readers will be drawn to the qualitative analysis, and they can make their own judgments as if just a patient-simulated analysis about clinical skills with helpful qualitative responses vs. an exercise that does increase PIF.